# Online Learning with Primary and Secondary Losses

**Avrim Blum**
Toyota Technological Institute at Chicago
avrim@ttic.edu

**Han Shao**
Toyota Technological Institute at Chicago
han@ttic.edu

## Abstract

We study the problem of online learning with primary and secondary losses. For example, a recruiter making decisions of which job applicants to hire might weigh false positives and false negatives equally (the primary loss) but the applicants might weigh false negatives much higher (the secondary loss). We consider the following question: Can we combine "expert advice" to achieve low regret with respect to the primary loss, while at the same time performing *not much worse than the worst expert* with respect to the secondary loss? Unfortunately, we show that this goal is unachievable without any bounded variance assumption on the secondary loss. More generally, we consider the goal of minimizing the regret with respect to the primary loss and bounding the secondary loss by a linear threshold. On the positive side, we show that running any switching-limited algorithm can achieve this goal if all experts satisfy the assumption that the secondary loss does not exceed the linear threshold by $o(T)$ for any time interval. If not all experts satisfy this assumption, our algorithms can achieve this goal given access to some external oracles which determine when to deactivate and reactivate experts.

## 1 Introduction

The online learning problem has been studied extensively in the literature and used increasingly in many applications including hiring, advertising and recommender systems. One classical problem in online learning is prediction with expert advice, in which a decision maker makes a sequence of $T$ decisions with access to $K$ strategies (also called "experts"). At each time step, the decision maker observes a scalar-valued loss of each expert. The standard objective is to perform as well as the best expert in hindsight. For example, a recruiter (the decision maker) sequentially decides which job applicants to hire with the objective of minimizing errors (of hiring an unqualified applicant and rejecting a qualified one). However, this may give rise to some social concerns since the decision receiver has a different objective (getting a job) which does not receive any attention. This problem can be modeled as an online learning problem with the primary loss (for the decision maker) and secondary loss (for the decision receiver). Taking the social impact into consideration, we ask the following question:

> *Can we achieve low regret with respect to the primary loss, while performing*
> *not much worse than the worst expert with respect to the secondary loss?*

Unfortunately, we answer this question negatively. More generally, we consider a bicriteria goal of minimizing the regret to the best expert with respect to the primary loss while minimizing the regret to a linear threshold $cT$ with respect to the secondary loss for some $c$. When the value of $c$ is set to the average secondary loss of the worst expert with respect to the secondary loss, the objective reduces to no-regret for the primary loss while performing no worse than the worst expert with respect to the secondary loss. Other examples, e.g., the average secondary loss of the worst expert with respect to the secondary loss among the experts with optimal primary loss, lead to different criteria of the secondary loss. Therefore, with the notion of regret to the linear threshold, we are able to study a more general goal. Based on this goal, we pose the following two questions:

1. If all experts have secondary losses no greater than $cT + o(T)$ for some $c$, can we achieve no-regret (compete comparably to the best expert) for the primary loss while achieving secondary loss no worse than $cT + o(T)$?

2. If we are given some external oracles to deactivate some "bad" experts with unsatisfactory secondary loss, can we perform as well as each expert with respect to the primary loss during the time they are active while achieving secondary loss no worse than $cT + o(T)$?

These two questions are trivial in the i.i.d. setting as we can learn the best expert with respect to the primary loss within $O(\log(T))$ rounds and then we just need to follow the best expert. In this paper, we focus on answering these two questions in the adversarial online setting.

## 1.1 Contributions

**An impossibility result without a bounded variance assumption**   We show that without any constraints on the variance of the secondary loss, even if all experts have secondary loss no greater than $cT$, achieving no-regret with respect to the primary loss and bounding secondary loss by $cT + O(T)$ is still unachievable. This answers our motivation question that it is impossible to achieve low regret with respect to the primary loss, while performing not much worse than the worst expert with respect to the secondary loss. This result explains why minimizing one loss while bounding another is non-trivial and applying existing algorithms for scalar-valued losses after scalarizing primary and secondary losses does not work. We propose an assumption on experts that the secondary loss of the expert during any time interval does not exceed $cT$ by $O(T^\alpha)$ for some $\alpha \in [0, 1)$.

Then we study the problem in two scenarios, a "good" one in which all experts satisfy this assumption and a "bad" one in which experts partially satisfy this assumption and we are given access to an external oracle to deactivate and reactivate experts.

**Our results in the "good" scenario**   In the "good" scenario, we show that running an algorithm with limited switching rounds such as Follow the Lazy Leader [Kalai and Vempala, 2005] and Shrinking Dartboard (SD) [Geulen et al., 2010] can achieve both regret to the best with respect to the primary loss and regret to $cT$ with respect to the secondary loss at $O(T^{\frac{1+\alpha}{2}})$. We also provide a lower bound of $\Omega(T^\alpha)$.

From another perspective, we relax the "good" scenario constraint by introducing adaptiveness to the secondary loss and constraining the variance of the secondary loss between any two switchings for any algorithm instead of that of any expert. We show that in this weaker version of "good" scenario, the upper bound of running switching-limited algorithms matches the lower bound at $\Theta(T^{\frac{1+\alpha}{2}})$.

**Our results in the "bad" scenario**   In the "bad" scenario, we assume that we are given an external oracle to determine which experts to deactivate as they do not satisfy the bounded variance assumption. We study two oracles here. One oracle deactivates the experts which do not satisfy the bounded variance assumption once detecting and never reactivates them. The other one reactivates those inactive experts at fixed rounds. In this framework, we are limited to select among the active experts at each round and we adopt a more general metric, sleeping regret, to measure the performance of the primary loss. We provide algorithms for the two oracles with theoretical guarantees on the sleeping regrets with respect to the primary loss and the regret to $cT$ with respect to the secondary loss.

## 1.2 Related work

One line of closely related work is online learning with multi-objective criterion. A bicriteria setting which examines not only the regret to the best expert but also the regret to a fixed mixture of all experts is investigated by Even-Dar et al. [2008], Kapralov and Panigrahy [2011], Sani et al. [2014]. The objective by Even-Dar et al. [2009] is to learn an optimal static allocation over experts with respect to a global cost function. Another multi-objective criterion called the Pareto regret frontier studied by Koolen [2013] examines the regret to each expert. Different from our work, all these criteria are studied in the setting of scalar-valued losses. The problem of multiple loss functions is studied by Chernov and Vovk [2009] under a heavy geometric restriction on loss functions. For vector losses, one fundamental concept is the Pareto front, the set of feasible points in which none can be dominated by any other point given several criteria to be optimized [Hwang and Masud, 2012, Auer et al., 2016]. However, the Pareto front contains unsatisfactory solutions such as the one minimizing the secondary loss, which implies that learning the Pareto front can not achieve our goal. Another classical concept is approachability, in which a learner aims at making the averaged vector loss converge to a pre-specified target set [Blackwell et al., 1956, Abernethy et al., 2011]. However,

we show that our fair solution is unapproachable without additional bounded variance assumptions. Approachability to an expansion target set based on the losses in hindsight is studied by Mannor et al. [2014]. However, the expansion target set is not guaranteed to be meet our criteria. Multi-objective criterion has also been studied in multi-armed bandits [Turgay et al., 2018].

## 2 Model

We consider the adversarial online learning setting with a set of $K$ experts $\mathcal{H} = \{1, \ldots, K\}$. At round $t = 1, 2, \ldots, T$, given an active expert set $\mathcal{H}_t \subseteq \mathcal{H}$, an online learner $\mathcal{A}$ computes a probability distribution $p_t \in \Delta_K$ over $\mathcal{H}$ with support only over $\mathcal{H}_t$ and selects one expert from $p_t$. Simultaneously an adversary selects two loss vectors $\ell_t^{(1)}, \ell_t^{(2)} \in [0,1]^K$, where $\ell_{t,h}^{(1)}$ and $\ell_{t,h}^{(2)}$ are the primary and secondary losses of expert $h \in \mathcal{H}$ at time $t$. Then $\mathcal{A}$ observes the loss vector and incurs expected losses $\ell_{t,\mathcal{A}}^{(i)} = p_t^\top \ell_t^{(i)}$ for $i \in \{1,2\}$. Let $L_{T,h}^{(i)} = \sum_{t=1}^T \ell_{t,h}^{(i)}$ denote the loss of expert $h$ and $L_{T,\mathcal{A}}^{(i)} = \sum_{t=1}^T p_t^\top \ell_t^{(i)}$ denote the loss of algorithm $\mathcal{A}$ for $i \in \{1,2\}$ during the first $T$ rounds. We will begin by focusing on the case that the active expert set $\mathcal{H}_t = \mathcal{H}$.

### 2.1 Regret notions

Traditionally, the regret (to the best) is used to measure the scalar-valued loss performance of a learner, which compares the loss of the learner and the best expert in hindsight. Similar to Even-Dar et al. [2008], we adopt the regret notion of $\mathcal{A}$ with respect to the primary loss as

$$\text{Reg}^{(1)} \triangleq \max\left(L_{T,\mathcal{A}}^{(1)} - \min_{h \in \mathcal{H}} L_{T,h}^{(1)}, 1\right).$$

We introduce another metric for the secondary loss called *regret to cT* for some $c \in [0,1]$, which compares the secondary loss of the learner with a linear term $cT$,

$$\text{Reg}_c^{(2)} \triangleq \max\left(L_{T,\mathcal{A}}^{(2)} - cT, 1\right).$$

Sleeping experts are developed to model the problem in which not all experts are available at all times [Blum, 1997, Freund et al., 1997]. At each round, each expert $h \in \mathcal{H}$ decides to be active or not and then a learner can only select among the active experts, i.e. have non-zero probability $p_{t,h}$ over the active experts. The goal is to perform as well as $h^*$ in the rounds where $h^*$ is active for all $h^* \in \mathcal{H}$. We denote by $\mathcal{H}_t$ the set of active experts at round $t$. The sleeping regret for the primary loss with respect to expert $h^*$ is defined as

$$\text{SleepReg}^{(1)}(h^*) \triangleq \max\left(\sum_{t:h^* \in \mathcal{H}_t} \sum_{h \in \mathcal{H}_t} p_{t,h} \ell_{t,h}^{(1)} - \sum_{t:h^* \in \mathcal{H}_t} \ell_{t,h^*}^{(1)}, 1\right).$$

The sleeping regret notion we adopt here is different from the regret to the best ordering of experts in the sleeping expert setting of Kleinberg et al. [2010]. Since achieving the optimal regret bound in Kleinberg's setting is computationally hard [Kanade and Steinke, 2014], we focus on the sleeping regret notion defined above.

### 2.2 Assumptions

Following a standard terminology, we call an adversary oblivious if her selection is independent of the learner's actions. Otherwise, we call the adversary adaptive. First, we assume that the primary loss is oblivious. This is a common assumption in the online learning literature and this assumption holds throughout the paper.

**Assumption 1.** *The primary losses $\{\ell_t^{(1)}\}_{t \in [T]}$ are oblivious.*

For an expert $h \in \mathcal{H}$, we propose a bounded variance assumption on her secondary loss: the average secondary loss for any interval does not exceed $c$ much. More formally, the assumption is described as below.

**Assumption 2.** *For some given $c, \delta, \alpha \in [0,1]$ and for all expert $h \in \mathcal{H}$, for any $T_1, T_2 \in [T]$ with $T_1 \leq T_2$,*

$$\sum_{t=T_1}^{T_2} (\ell_{t,h}^{(2)} - c) \leq \delta T^\alpha.$$

We show that such a bounded variance assumption is necessary in Section 3. We call a scenario "good" if all experts satisfy assumption 2. Otherwise, we call the scenario "bad". This "good" constraint can be relaxed by introducing adaptiveness to the secondary loss. We have a relaxed version of the "good" scenario in which the average secondary loss between any two switchings does not exceed $c$ much for any algorithm. More formally,

**Assumption 2′.** *For some given $c, \delta, \alpha \in [0, 1]$, for any algorithm $\mathcal{A}$, let $\mathcal{A}_t \in \mathcal{H}$ denote the selected expert at round $t$. For any expert $h \in \mathcal{H}$ and $T_1 \in [T]$ such that $\mathcal{A}_{T_1} = h$ and $\mathcal{A}_{T_1-1} \neq h$ (where $\mathcal{A}_{T+1} = T \mathcal{A}_0 = 0$ for notation simplicity), we have*

$$\sum_{\tau=T_1}^{\min_{t > T_1 : \mathcal{A}_t \neq h} t - 1} \left( \ell^{(2)}_{\tau,h} - c \right) \leq \delta T^\alpha .$$

In the "good" scenario, the active expert set $\mathcal{H}_t = \mathcal{H}$ for all rounds and the goal is minimizing both $\mathrm{Reg}^{(1)}$ and $\mathrm{Reg}^{(2)}_c$. In the "bad" scenario, we consider that we are given an oracle which determines $\mathcal{H}_t$ at each round and the goal is minimizing $\mathrm{SleepReg}^{(1)}(h^*)$ for all $h^* \in \mathcal{H}$ and $\mathrm{Reg}^{(2)}_c$.

## 3  Impossibility result without any bounded variance assumption

In this section, we show that without any additional assumption on the secondary loss, even if all experts have secondary loss no greater than $cT$ for some $c \in [0, 1]$, there exists an adversary such that any algorithm incurs $\mathbb{E}[\max(\mathrm{Reg}^{(1)}, \mathrm{Reg}^{(2)}_c)] = \Omega(T)$.

**Theorem 1.** *Given a fixed expert set $\mathcal{H}$, there exists an adversary such that any algorithm will incur $\mathbb{E}[\max(\mathrm{Reg}^{(1)}, \mathrm{Reg}^{(2)}_c)] = \Omega(T)$ with $c = \max_{h \in \mathcal{H}} L^{(2)}_{T,h}/T$, where the expectation is taken over the randomness of the adversary.*

*Proof.* To prove this theorem, we construct a binary classification example as below.

In a binary classification problem, for each sample with true label $y \in \{+, -\}$ and prediction $\widehat{y} \in \{+, -\}$, the primary loss is defined as the expected $0/1$ loss for incorrect prediction, i.e., $\mathbb{E}_{y,\widehat{y}}\left[\mathbb{1}_{\{\widehat{y} \neq y\}}\right]$ and the secondary loss is defined as the expected $0/1$ loss for false negatives, i.e., $\mathbb{E}_{y,\widehat{y}}\left[\mathbb{1}_{\{\widehat{y} \neq y, y=+\}}\right]$. We denote by $h(b)$ the expert predicting $-$ with probability $b$ and $+$ otherwise. Then every expert can be represented by a sequence of values of $b$. At round $t$, the true label is negative with probability $a$. We divide $T$ into two phases evenly, $\{1, \ldots, T/2\}$ and $T/2 + 1, \ldots, T$, in each of which the adversary generates outcomes with different values of $a$ and two experts $\mathcal{H} = \{h_1, h_2\}$ have different values of $b$ in different phases. We construct two worlds with different values of $a$ and $b$ in phase 2 and any algorithm should have the same behavior in phase 1 of both worlds. The adversary randomly chooses one world with equal probability. The specific values of $a$ and $b$ are given in Table 1. Let $c = 1/16$.

Table 1: The values of $a$ and $b$ in different phases for the binary classification example.

| experts\phase | $1 : a = \frac{5}{8}$ | $2 : a = \frac{3}{4}$ (world I) | $2 : a = \frac{5}{8}$ (world II) |
|---|---|---|---|
| $h_1$ | $b = \frac{1}{6}$ | $b = 0$ | $b = \frac{1}{6}$ |
| $h_2$ | $b = 0$ | $b = \frac{1}{2}$ | $b = 0$ |

The loss of expert $h(b)$ is $\ell^{(1)}_{t,h(b)} = (1-a)b + a(1-b)$ and $\ell^{(2)}_{t,h(b)} = (1-a)b$. In phase 1 and phase 2 of world II, $\ell^{(1)}_{t,h_1} = 7/12$, $\ell^{(2)}_{t,h_1} = 1/16$, $\ell^{(1)}_{t,h_2} = 5/8$ and $\ell^{(2)}_{t,h_2} = 0$. In phase 2 of world I, $\ell^{(1)}_{t,h_1} = 3/4$, $\ell^{(2)}_{t,h_1} = 0$, $\ell^{(1)}_{t,h_2} = 1/2$ and $\ell^{(2)}_{t,h_2} = 1/8$. For any $h \in \mathcal{H}$, we have $L^{(2)}_{T,h} \leq T/16$.

For any algorithm which selects $h_1$ for $T_1$ (in expectation) rounds in phase 1 and $T_2$ (in expectation) rounds in phase 2 of world I. If $T_1 \leq T/4$, then $\mathrm{Reg}^{(1)} \geq (T/2 - T_1)/24 \geq T/96$ in world II; else if $T_1 > T/4$ and $T_2 \geq T_1/4$, then $\mathrm{Reg}^{(1)} \geq T_2/4 - T_1/24 \geq T/192$ in world I; else $\mathrm{Reg}^{(2)}_c = T_1/16 + (T/2 - T_2)/8 - T/16 = (T_1 - 2T_2)/16 \geq T/128$ in world I. In any case, we have $\mathbb{E}[\max(\mathrm{Reg}^{(1)}, \mathrm{Reg}^{(2)}_c)] = \Omega(T)$. $\qquad\square$

The proof of Theorem 1 implies that an expert with total secondary loss no greater than $cT$ but high secondary loss at the beginning will consume a lot of budget for secondary loss, which makes switching to other experts with low primary loss later costly in terms of secondary loss. The theorem answers our first question negatively, i.e., we are unable to achieve no-regret for primary loss while performing as well as the worst expert with respect to the secondary loss.

# 4    Results in the "good" scenario

In this section, we consider the problem of minimizing $\max(\mathrm{Reg}^{(1)}, \mathrm{Reg}_c^{(2)})$ with Assumption 2 or 2'. We first provide lower bounds of $\Omega(T^\alpha)$ under Assumption 2 and of $\Omega(T^{\frac{1+\alpha}{2}})$ under Assumption 2'. Then we show that applying any switching-limited algorithms such as Shrinking Dartboard (SD) [Geulen et al., 2010] and Follow the Lazy Leader (FLL) [Kalai and Vempala, 2005] can achieve $\max(\mathrm{Reg}^{(1)}, \mathrm{Reg}_c^{(2)}) = O(T^{\frac{1+\alpha}{2}})$ under Assumption 2 or 2', which matches the lower bound under Assumption 2'.

## 4.1    Lower bound

**Theorem 2.** *If Assumption 2 holds with some given $c, \delta, \alpha$, then there exists an adversary such that any algorithm incurs $\mathbb{E}[\max(\mathrm{Reg}^{(1)}, \mathrm{Reg}_c^{(2)})] = \Omega(T^\alpha)$.*

*Proof.* We construct a binary classification example to prove the lower bound.

The losses and the experts $\mathcal{H} = \{h_1, h_2\}$ are defined based on $h(b)$ in the same way as that in the proof of Theorem 1. We divide $T$ into 3 phases, the first two of which have $T^\alpha$ rounds and the third has $T - 2T^\alpha$ rounds. Each expert has different $b$s in different phases as shown in Table 2. At each time $t$, the sample is negative with probability $3/4$. We set $c = 0$.

Since $(\ell_{t,h(0)}^{(1)}, \ell_{t,h(0)}^{(2)}) = (3/4, 0)$ and $(\ell_{t,h(1)}^{(1)}, \ell_{t,h(1)}^{(2)}) = (1/4, 1/4)$, the cumulative loss for both experts are $(L_{T,h}^{(1)}, L_{T,h}^{(2)}) = (3T/4 - T^\alpha/2, T^\alpha/4)$. Any algorithm $\mathcal{A}$ achieving $L_{T,h}^{(1)} \le 3T/4 - T^\alpha/4$ will incur $\mathrm{Reg}_c^{(2)} \ge T^\alpha/8$. $\qquad\square$

Table 2: The values of $b$ in different phases for the binary classification example.

| experts\phase | $1 : T^\alpha$ | $2 : T^\alpha$ | $3 : T - 2T^\alpha$ |
|---|---|---|---|
| $h_1$ | $b = 1$ | $b = 0$ | $b = 0$ |
| $h_2$ | $b = 0$ | $b = 1$ | $b = 0$ |

Combined with the classical lower bound of $\Omega(\sqrt{T})$ in online learning [Cesa-Bianchi and Lugosi, 2006], $\mathbb{E}[\max(\mathrm{Reg}^{(1)}, \mathrm{Reg}_c^{(2)})] = \Omega(\max(T^\alpha, \sqrt{T}))$. In the relaxed version of the "good" scenario, we have the following theorem.

**Theorem 3.** *If Assumption 2' holds with some given $c, \delta, \alpha$, then there exists an adversary such that any algorithm incurs $\mathbb{E}[\max(\mathrm{Reg}^{(1)}, \mathrm{Reg}_c^{(2)})] = \Omega(T^{\frac{1+\alpha}{2}})$.*

**Sketch of the proof**    Inspired by the proof of the lower bound by Altschuler and Talwar [2018], we construct an adversary such that any algorithm achieving $\mathrm{Reg}^{(1)} = O(T^{\frac{1+\alpha}{2}})$ has to switch for some number of times. For the secondary loss, the adversary sets $\ell_{t,h}^{(2)} = c$ only if $h$ has been selected for more than $T^\alpha$ rounds consecutively until time $t - 1$; otherwise $\ell_{t,h}^{(2)} = c + \delta$. In this case, every switching will increase the secondary loss. Then we can show that either $\mathrm{Reg}^{(1)}$ or $\mathrm{Reg}_c^{(2)}$ is $\Omega(T^{\frac{1+\alpha}{2}})$. The complete proof can be found in Appendix A.

## 4.2    Algorithm

Under Assumption 2 or 2', we are likely to suffer an extra $\delta T^\alpha$ secondary loss every time we switch from one expert to another. Inspired by this, we can upper bound $\max(\mathrm{Reg}^{(1)}, \mathrm{Reg}_c^{(2)})$ by limiting the number of switching times. Given a switching-limited learner $\mathcal{L}$ on scalar-valued losses, e.g., Shrinking Dartboard (SD) [Geulen et al., 2010] and Follow the Lazy Leader (FLL) [Kalai and Vempala, 2005], our algorithm $\mathcal{A}_{\mathrm{SL}}(\mathcal{L})$ is described as below.

We divide the time horizon into $T^{1-\alpha}$ epochs evenly and within each epoch we select the same expert. Let $e_i = \{(i-1)T^\alpha + 1, \ldots, iT^\alpha\}$ denote the $i$-th epoch and $\ell_{e_i,h}^{(1)} = \sum_{t \in e_i} \ell_{t,h}^{(1)}/T^\alpha$ denote the average primary loss of the $i$-th epoch. We apply $\mathcal{L}$ over $\{\ell_{e_i,h}^{(1)}\}_{h \in \mathcal{H}}$ for $i = 1, \ldots, T^{1-\alpha}$. Let $s_{\mathrm{SL}}(E)$ and $r_{\mathrm{SL}}(E)$ denote the expected number of switching times and the regret of running $\mathcal{L}$ for $E$ rounds. Then we have the following theorem.

**Theorem 4.** *Under Assumption 2 or 2′, given a switching-limited learner $\mathcal{L}$, $\mathcal{A}_{\mathrm{SL}}(\mathcal{L})$ achieves* $\mathrm{Reg}^{(1)} \leq T^\alpha r_{\mathrm{SL}}(T^{1-\alpha})$ *and* $\mathrm{Reg}_c^{(2)} \leq \delta T^\alpha (s_{\mathrm{SL}}(T^{1-\alpha}) + 1)$. *By adopting SD or FLL as the learner $\mathcal{L}$, $\mathcal{A}_{\mathrm{SL}}(\mathrm{SD})$ and $\mathcal{A}_{\mathrm{SL}}(\mathrm{FLL})$ achieve* $\max(\mathrm{Reg}^{(1)}, \mathrm{Reg}_c^{(2)}) = O(\sqrt{\log(K)T^{1+\alpha}})$.

*Proof.* It is obvious that $\mathrm{Reg}^{(1)} \leq T^\alpha r_{\mathrm{SL}}(T^{1-\alpha})$. We denote by $S$ the random variable of the total number of switching times and $\tau_1, \ldots, \tau_S$ the time steps the algorithm switches. For notation simplicity, let $\tau_0 = 1$ and $\tau_{S+1} = T + 1$. Then $\mathrm{Reg}_c^{(2)} = \mathbb{E}_\mathcal{A}[\sum_{t=1}^T (\ell_{t,\mathcal{A}_t}^{(2)} - c)] \leq \mathbb{E}_\mathcal{A}[\sum_{s=0}^S \sum_{t=\tau_s}^{\tau_{s+1}-1} (\ell_{t,\mathcal{A}_t}^{(2)} - c)] \leq \mathbb{E}_\mathcal{A}[\sum_{s=0}^S \delta T^\alpha] = \delta T^\alpha (s_{\mathrm{SL}}(T^{1-\alpha}) + 1)$. Both SD and FLL have $s_{\mathrm{SL}}(T^{1-\alpha}) = O(\sqrt{\log(K)T^{1-\alpha}})$ and $r_{\mathrm{SL}}(T^{1-\alpha}) = O(\sqrt{\log(K)T^{1-\alpha}})$ [Geulen et al., 2010, Kalai and Vempala, 2005], which completes the proof. $\square$

$\mathcal{A}_{\mathrm{SL}}(\mathrm{SD})$ and $\mathcal{A}_{\mathrm{SL}}(\mathrm{FLL})$ match the lower bound at $\Theta(T^{\frac{1+\alpha}{2}})$ under Assumption 2′. But there is a gap between the upper bound $O(T^{\frac{1+\alpha}{2}})$ and the lower bound $\Omega(T^\alpha)$ under Assumption 2, which is left as an open question. We investigate this question a little bit by answering negatively if the analysis of $\mathcal{A}_{\mathrm{SL}}(\mathcal{L})$ can be improved to achieve $O(T^\alpha)$. We define a class of algorithms which depends only on the cumulative losses of the experts, i.e., there exists a function $g : \mathbb{R}^{2K} \mapsto \Delta^K$ such that $p_t = g(L_{t-1}^{(1)}, L_{t-1}^{(2)})$. Many classical algorithms such as Exponential Weights [Littlestone et al., 1989] and Follow the Perturbed Leader [Kalai and Vempala, 2005] are examples in this class. The following theorem show that any algorithm dependent only on the cumulative losses cannot achieve a better bound than $\Omega(T^{\frac{1+\alpha}{2}})$, which provides some intuition on designing algorithms for future work. The detailed proof can be found in Appendix B.

**Theorem 5.** *Under Assumption 2, for any algorithm only dependent on the cumulative losses of the experts,* $\mathbb{E}[\max(\mathrm{Reg}^{(1)}, \mathrm{Reg}_c^{(2)})] = \Omega(T^{\frac{1+\alpha}{2}})$.

## 5 Results in the "bad" scenario

In the "bad" scenario, some experts may have secondary losses with high variance. To compete with the best expert in the period in which it has low variance, we assume that the learner is given some fixed external oracle determining which experts to deactivate and reactivate. In this section, we consider the goal of minimizing $\mathrm{SleepReg}^{(1)}(h^*)$ for all $h^* \in \mathcal{H}$ and $\mathrm{Reg}_c^{(2)}$. Here we study two oracles: one deactivates the "unsatisfactory" expert if detecting high variance of the secondary loss and never reactivates it again; the other one deactivates the "unsatisfactory" expert if detecting high variance of the secondary loss and reactivates it at fixed time steps.

### 5.1 The first oracle: deactivating the "unsatisfactory" experts

The oracle is described as below. The active expert set is initialized to contain all experts $\mathcal{H}_1 = \mathcal{H}$. At time $t = 1, \ldots, T$, we let $\Delta \mathcal{H}_t = \{h \in \mathcal{H}_t : \exists t' \leq t, \sum_{\tau=t'}^t (\ell_{\tau,h}^{(2)} - c) > \delta T^\alpha\}$ denote the set of active experts which do not satisfy Assumption 2. Then we remove these experts from the active expert set, i.e., $\mathcal{H}_{t+1} = \mathcal{H}_t \setminus \Delta \mathcal{H}_t$. We assume that there always exist some active experts, i.e. $\mathcal{H}_T \neq \emptyset$.

One direct way is running $\mathcal{A}_{\mathrm{SL}}(\mathcal{L})$ as a subroutine and restarting $\mathcal{A}_{\mathrm{SL}}(\mathcal{L})$ at time $t$ if there exist experts deactivated at the end of $t-1$, i.e., $\Delta H_{t-1} \neq \emptyset$. However, restarting will lead to linear dependency on $K$ for sleeping regrets. To avoid this linear dependency, we construct pseudo primary losses for each expert such that if $h$ is active at time $t$, $\widetilde{\ell}_{t,h}^{(1)} = \ell_{t,h}^{(1)}$; otherwise, $\widetilde{\ell}_{t,h}^{(1)} = 1$. The probability of selecting inactive experts degenerates due to the high pseudo losses. For those inactive experts we cannot select, we construct a mapping $f : \mathcal{H} \mapsto \mathcal{H}$, which maps each expert to an active expert. If $\mathcal{A}_{\mathrm{SL}}(\mathcal{L})$ decides to select an inactive expert $h$ at time $t$, we will select $f(h)$ instead. The detailed algorithm is described in Algorithm 1. Although the algorithm takes $\alpha$ as an input, it is worth to mention that the algorithm only uses $\alpha$ to decide the length of each epoch. We can choose a different epoch length and derive different regret upper bounds.

---
**Algorithm 1** $\mathcal{A}_1$
---
1: **Input:** $T$, $\mathcal{H}$, $\alpha$ and a learner $\mathcal{L}$
2: Initialize $f(h) = h$ for all $h \in \mathcal{H}$.
3: Start an instance $\mathcal{A}_{\mathrm{SL}}(\mathcal{L})$.
4: **for** $t = 1, \ldots, T$ **do**
5:   Get expert $h_t$ from $\mathcal{A}_{\mathrm{SL}}(\mathcal{L})$.
6:   Select expert $f(h_t)$.
7:   Feed $\widetilde{\ell}_t^{(1)}$ to $\mathcal{A}_{\mathrm{SL}}(\mathcal{L})$.
8:   For all $h$ with $f(h) \in \Delta\mathcal{H}_t$, set $f(h) = h_0$, where $h_0$ is any expert in $\mathcal{H}_{t+1}$.
9: **end for**
---

**Theorem 6.** *Let $T_{h^*}$ denote the number of rounds where expert $h^*$ is active. Running Algorithm 1 with learner $\mathcal{L}$ being SD or FLL can achieve*

$$\mathrm{SleepReg}^{(1)}(h^*) = O(\sqrt{\log(K) T_{h^*} T^\alpha}), \tag{1}$$

*for all $h^* \in \mathcal{H}$ and*

$$\mathrm{Reg}_c^{(2)} = O(\sqrt{\log(K) T^{1+\alpha}} + KT^\alpha). \tag{2}$$

*Proof.* Since $\ell_{m,h}^{(1)} \leq \widetilde{\ell}_{m,h}^{(1)}$, we have

$$\mathrm{SleepReg}^{(1)}(h^*) = \left( \sum_{t=1}^{T_{h^*}} \mathbb{E}_\mathcal{A}\left[ \ell_{t,\mathcal{A}_t}^{(1)} \right] - \sum_{t=1}^{T_{h^*}} \ell_{t,h^*}^{(1)} \right) \leq \left( \sum_{t=1}^{T_{h^*}} \mathbb{E}_\mathcal{A}\left[ \widetilde{\ell}_{t,\mathcal{A}_t}^{(1)} \right] - \sum_{t=1}^{T_{h^*}} \widetilde{\ell}_{t,h^*}^{(1)} \right)$$
$$= O(\sqrt{\log(K) T_{h^*} T^\alpha}),$$

where the last step uses the results in Theorem 4. It is quite direct to have $\mathrm{Reg}_c^{(2)} = O(\delta T^\alpha(\sqrt{\log(K) T^{1-\alpha}} + K)) = O(\sqrt{\log(K) T^{1+\alpha}} + KT^\alpha)$, where the first term comes from the number of switching times for running $\mathcal{A}_{\mathrm{SL}}$ and the second term comes from an extra switching caused by deactivating one expert. $\square$

For the sleeping regret for expert $h^*$, the right hand side in Eq. (1) is $o(T_{h^*})$ if $T_{h^*} = \omega(T^\alpha)$, which is consistent with the impossibility result without bounded variance in Section 3. When $\alpha \geq 1/2$, the right hand side of Eq. (2) is dominated by $KT^\alpha$. This linear dependency on $K$ is inevitable if we want to have $\mathrm{SleepReg}_{h^*}^{(1)} = o(T_{h^*})$ for all $h^* \in \mathcal{H}$. The proof is given in Appendix C.

**Theorem 7.** *Let $T_{h^*} = \omega(T^\alpha)$ for all $h^* \in \mathcal{H}$. There exists an adversary such that any algorithm achieving $\mathrm{SleepReg}_{h^*}^{(1)} = o(T_{h^*})$ for all $h^* \in \mathcal{H}$ will incur $\mathrm{Reg}_c^{(2)} = \Omega(KT^\alpha)$ for $K = O(\log(T))$.*

### 5.2 The second oracle: reactivating at fixed times

Now we consider the oracle which deactivates the unsatisfactory experts once detecting and reactivate them at fixed times. The oracle is described as follows. At given $N + 1$ fixed time steps $t_0 = 1, t_1, \ldots, t_N$ with $t_{n+1} - t_n = \Omega(T^\beta)$ for some $\beta > \alpha$ (where $t_{N+1} = T + 1$ for notation simplicity), the active expert set $\mathcal{H}_t$ is reset to $\mathcal{H}$. At time $t = t_n, \ldots, t_{n+1} - 2$ for any $n = 0, \ldots, N$, the experts $\Delta\mathcal{H}_t = \{ h \in \mathcal{H}_t : \exists t' \text{ such that } t_n \leq t' \leq t, \sum_{\tau=t'}^t (\ell_{\tau,h}^{(2)} - c) > \delta T^\alpha \}$ will be deactivated, i.e. $\mathcal{H}_{t+1} = \mathcal{H}_t \setminus \Delta\mathcal{H}_t$. We assume that there always exists some satisfactory experts, i.e. $\mathcal{H}_{t_n-1} \neq \emptyset$ for all $n = 1, \ldots, N + 1$.

Restarting Algorithm 1 at $t = t_0, \ldots, t_N$ is one of the most direct methods. Let $T_{h^*}^{(n)}$ denote the number of rounds $h^*$ is active during $t = t_n, \ldots, t_{n+1} - 1$ and $T_{h^*} = \sum_{n=0}^N T_{h^*}^{(n)}$ denote the total number of rounds $h^*$ is active. Then we have $\mathrm{SleepReg}_{h^*}^{(1)} = O(\sum_{n=0}^N \sqrt{\log(K) T_{h^*}^{(n)} T^\alpha}) = O(\sqrt{\log(K) T_{h^*} T^\alpha N})$ and $\mathrm{Reg}_c^{(2)} = O(\sum_{n=0}^N (\sqrt{\log(K) T^\alpha (t_{n+1} - t_n)} + K\delta T^\alpha)) = O(\sqrt{\log(K) T^{1+\alpha} N} + NKT^\alpha)$.

However, if all experts are active all times, then the upper bound of $\mathrm{SleepReg}^{(1)}(h^*)$ for the algorithm of restarting is $O(\sqrt{\log(K) T^{1+\alpha} N}) = O(\sqrt{\log(K) T^{2+\alpha-\beta}})$, which is quite large. We consider

a smarter algorithm with better sleeping regrets when $T_{h^*}$ is large. The algorithm combines the methods of constructing meta experts for time-selection functions by Blum and Mansour [2007] to bound the sleeping regrets and inside each interval, we select experts based on SD [Geulen et al., 2010] to bound the number of switching times. We run the algorithm in epochs with length $T^\alpha$ and within each epoch we play the same expert. For simplicity, we assume that the active expert set will be updated only at the beginning of each epoch, which can be easily generalized. Let $e_i = \{(i-1)T^\alpha + 1, \ldots, iT^\alpha\}$ denote the $i$-th epoch and $E = \{e_i\}_{i \in [T^{1-\alpha}]}$ denote the set of epochs. We let $\ell_{e,h}^{(1)} = \sum_{t \in e} \ell_{t,h}^{(1)}/T^\alpha$ and $\ell_{e,\mathcal{A}}^{(1)} = \sum_{t \in e} \ell_{t,\mathcal{A}_t}^{(1)}/T^\alpha$ denote the average primary loss of expert $h$ and the algorithm. And we let $\mathcal{H}_e$ and $\Delta\mathcal{H}_e$ denote the active expert set at the beginning of epoch $e$ and the deactivated expert set at the end of epoch $e$. Then we define the time selection function for epoch $e$ as $I_{h^*}(e) = \mathbb{1}(h^*$ is active in epoch $e)$ for each $h^* \in \mathcal{H}$. Then we construct $K$ meta experts for each time selection function. Similar to Algorithm 1, we adopt the same expert mapping function $f$ and using pseudo losses $\widetilde{\ell}_{e,h}^{(1)} = \ell_{e,h}^{(1)}$ if $h$ is active and $\widetilde{\ell}_{e,h}^{(1)} = 1$ if not. The detailed algorithm is shown as Algorithm 2. Then we have the following theorem, the detailed proof of which is provided in Appendix D.

**Theorem 8.** *Running Algorithm 2 can achieve*

$$\text{SleepReg}^{(1)}(h^*) = O(\sqrt{\log(K)T^{1+\alpha}} + T_{h^*}\sqrt{\log(K)T^{\alpha-1}}) ,$$

*for all $h^* \in \mathcal{H}$ and*

$$\text{Reg}_c^{(2)} = O(\sqrt{\log(K)T^{1+\alpha}} + \log(K)T^\alpha N + NKT^\alpha) .$$

Algorithm 2 achieves $o(T_{h^*})$ sleeping regrets for $h^*$ with $T_{h^*} = \omega(T^{\frac{1+\alpha}{2}})$ and outperforms restarting Algorithm 1 when $NT_{h^*} = \omega(T)$. $\text{SleepReg}^{(1)}(h^*)$ of Algorithm 2 is $O(\sqrt{\log(K)T_{h^*}^{1+\alpha}})$ when $T_{h^*} = \Theta(T)$, which matches the results in Theorem 4.

---

**Algorithm 2** $\mathcal{A}_2$
---
1: **Input:** $T$, $\mathcal{H}$, $\alpha$ and $\eta$
2: Initialize $f(h) = h$ for all $h \in \mathcal{H}$.
3: $w_{1,h}^{h^*} = \frac{1}{K}$ for all $h \in \mathcal{H}$, for all $h^* \in \mathcal{H}$.
4: **for** $m = 1, \ldots, T^{1-\alpha}$ **do**
5: $\quad w_{m,h} = \sum_{h^*} I_{h^*}(e_m)w_{m,h}^{h^*}$, $W_m = \sum_h w_{m,h}$ and $p_{m,h} = \frac{w_{m,h}}{W_m}$.
6: $\quad$ **if** $m \in \{(t_n-1)/T^{1-\alpha} + 1\}_{n=0}^N$ **then** get $h_m$ from $p_m$. **else**
7: $\quad\quad$ With prob. $\frac{w_{m,h_{m-1}}}{w_{m-1,h_{m-1}}}$, get $h_m = h_{m-1}$; with prob. $1 - \frac{w_{m,h_{m-1}}}{w_{m-1,h_{m-1}}}$, get $h_m$ from $p_m$.
8: $\quad$ **end if**
9: $\quad$ Select expert $f(h_m)$.
10: $\quad$ Update $w_{m+1,h}^{h^*} = w_{m,h}^{h^*}\eta^{I_{h^*}(e_m)(\widetilde{\ell}_{e_m,h}^{(1)} - \eta\widetilde{\ell}_{e_m,\mathcal{A}}^{(1)})+1}$ for all $h, h^* \in \mathcal{H}$.
11: $\quad$ For all $h$ with $f(h) \in \Delta\mathcal{H}_{e_m}$, set $f(h) = h_0$, where $h_0$ is any expert in $\mathcal{H}_{e_{m+1}}$.
12: **end for**
---

## 6 Discussion

We introduce the study of online learning with primary and secondary losses. We find that achieving no-regret with respect to the primary loss while performing no worse than the worst expert with respect to the secondary loss is impossible in general. We propose a bounded variance assumption over experts such that we can control secondary losses by limiting the number of switching times. Therefore, we are able to bound the regret with respect to the primary loss and the regret to $cT$ with respect to the secondary loss. Our work is only a first step in this problem and there are several open questions.

One is the optimality under Assumption 2. As aforementioned, our bounds of $\max(\text{Reg}^{(1)}, \text{Reg}_c^{(2)})$ in the "good" scenario are not tight and we show that any algorithm only dependent on the cumulative losses will have $\text{Reg}^{(1)} = \Omega(T^{\frac{1+\alpha}{2}})$, which indicates that the optimal algorithm cannot only depends on the cumulative losses if the optimal bound is $o(T^{\frac{1+\alpha}{2}})$. Under Assumption 2', the upper bound of the algorithm of limiting switching matches the lower bound. This possibly implies that limiting switching may not be the best way to make use of the information provided by Assumption 2.

In the "bad" scenario with access to the oracle which reactivates experts at fixed times, our sleeping regret bounds depend not only on $T_{h^*}$ but also on $T$, which makes the bounds meaningless when $T_{h^*}$ is small. It is unclear if we can obtain optimal sleeping regrets dependent only on $T_{h^*}$ for all $h^* \in \mathcal{H}$. The algorithm of *Adanormalhedge* by Luo and Schapire [2015] can achieve sleeping regret of $O(\sqrt{T_{h^*}})$ without bound on the number of switching actions. However, how to achieve sleeping regret of $o(T_{h^*})$ with limited switching cost is of independent research interest.

In the "bad" scenario where Assumption 2 does not hold, we assume that $c$ is pre-specified and known to the oracle. Theorem 1 show that achieving $\max(\text{Reg}^{(1)}, \text{Reg}_c^{(2)}) = o(T)$ with $c = \max_h L_{T,h}^{(2)}$ is impossible without any external oracle. How to define a setting an unknown $c$ and design a reasonable oracle in this setting is an open question.

## Broader Impact

This research studies a society-constrained online decision making problem, where we take the decision receiver's objective into consideration. Therefore, in a decision making process (e.g. deciding whether to hire a job applicant, whether to approve a loan, or whether to admit a student to an honors class), the decision receiver (e.g., job applicants, loan applicants, students) could benefit from our study at the cost of increasing the loss of the decision maker (e.g., recruiters, banks, universities) a little. The consequences of failure of the system and biases in the data are not applicable.

## Acknowledgments and Disclosure of Funding

This work was supported in part by the National Science Foundation under grant CCF-1815011.

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
