[Supplementary Material]

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

*Proof.* We construct an adversary with oblivious primary losses and adaptive secondary losses to prove the theorem. The adversary is inspired by the proof of the lower bound by Altschuler and Talwar [2018]. We divide $T$ into $T^{1-\alpha}$ epochs evenly and the primary losses do not change within each epoch. Let $\lceil t \rceil_e = \min_{m:mT^\alpha \geq t} mT^\alpha$ denote the last time step of the epoch containing time step $t$. For each expert $h \in \mathcal{H}$, at the beginning of each epoch, we toss a fair coin and let $\ell_{t,h}^{(1)} = 0$ if it is head and $\ell_{t,h}^{(1)} = 1$ if it is tail. It is well-known that there exists a universal constant $a$ such that $\mathbb{E}\left[\min_{h \in \mathcal{H}} Z_h\right] = E/2 - a\sqrt{E \log(K)}$ where $Z_h \sim \mathrm{Bin}(E, 1/2)$. Then we have

$$\mathbb{E}\left[\min_{h \in \mathcal{H}} \sum_{t=1}^{T} \ell_{t,h}^{(1)}\right] \leq \frac{T}{2} - aT^{\frac{1+\alpha}{2}}\sqrt{\log(K)} .$$

For algorithm $\mathcal{A}$, let $\mathcal{A}_t$ denote the selected expert at time $t$. Then we construct adaptive secondary losses as follows. First, for the first $T^\alpha$ rounds, $\ell_{t,h}^{(2)} = c + \delta$ for all $h \in \mathcal{H}$. For $t \geq T^\alpha + 1$,

$$\ell_{t,h}^{(2)} = \begin{cases} c & \text{if } h = \mathcal{A}_{t-1} = \ldots = \mathcal{A}_{t-T^\alpha} \\ c + \delta & \text{otherwise} \end{cases} .$$

This indicates that the algorithm can obtain $\ell_{t,\mathcal{A}_t}^{(2)} = c$ only by selecting the expert she has consecutively selected in the last $T^\alpha$ rounds and that each switching leads to $\ell_{t,\mathcal{A}_t}^{(2)} = c + \delta$. Let $S$ denote the total number of switchings and $\tau_1, \ldots, \tau_S$ denote the time steps $\mathcal{A}$ switches. For notation simplicity, let $\tau_{S+1} = T+1$. If $\mathbb{E}\left[L_{T,\mathcal{A}}^{(1)}\right] \geq T/2 - aT^{\frac{1+\alpha}{2}}\sqrt{\log(K)}/2$, then $\mathbb{E}\left[\mathrm{Reg}^{(1)}\right] \geq aT^{\frac{1+\alpha}{2}}\sqrt{\log(K)}/2$; otherwise,

$$\frac{T}{2} - \frac{1}{2}\mathbb{E}\left[\sum_{s=1}^{S} \min\left(\tau_{s+1} - \tau_s, \lceil \tau_s \rceil_e + 1 - \tau_s\right)\right] \overset{(a)}{\leq} \mathbb{E}\left[L_{T,\mathcal{A}}^{(1)}\right] < \frac{T}{2} - aT^{\frac{1+\alpha}{2}}\sqrt{\log(K)}/2 ,$$

where Eq. (a) holds due to that the $s$-th switching helps to decrease the expected primary loss by at most $\min\left(\tau_{s+1} - \tau_s, \lceil \tau_s \rceil_e + 1 - \tau_s\right)/2$. Since the $s$-th switching increases the secondary loss to $c + \delta$ for at least $\min(\tau_{s+1} - 1 - \tau_s, T^\alpha)$ rounds, then we have

$$\mathbb{E}\left[L_{T,\mathcal{A}}^{(2)}\right] \geq cT + \delta\mathbb{E}\left[\sum_{s=1}^{S} \min(\tau_{s+1} - \tau_s, T^\alpha)\right]$$

$$\geq cT + \delta\mathbb{E}\left[\sum_{s=1}^{S} \min\left(\tau_{s+1} - \tau_s, \lceil \tau_s \rceil_e + 1 - \tau_s\right)\right]$$

$$> cT + \delta aT^{\frac{1+\alpha}{2}}\sqrt{\log(K)},$$

which indicates that $\mathbb{E}\left[\mathrm{Reg}_c^{(2)}\right] = \Omega(T^{\frac{1+\alpha}{2}})$. Therefore, $\mathbb{E}\left[\max\left(\mathrm{Reg}^{(1)}, \mathrm{Reg}_c^{(2)}\right)\right] \geq \max\left(\mathbb{E}\left[\mathrm{Reg}^{(1)}\right], \mathbb{E}\left[\mathrm{Reg}_c^{(2)}\right]\right) = \Omega(T^{\frac{1+\alpha}{2}})$. $\square$

# B  Proof of Theorem 5

**Theorem 5.** *Under Assumption 2, for any algorithm only dependent on the cumulative losses of the experts, $\mathbb{E}[\max(\mathrm{Reg}^{(1)}, \mathrm{Reg}_c^{(2)})] = \Omega(T^{\frac{1+\alpha}{2}})$.*

*Proof.* We divide $T$ into $T^{1-\beta}$ intervals evenly with $\beta = \frac{1+\alpha}{2}$ and construct $T^{1-\beta} + 1$ worlds with 2 experts. For computation simplicity, we let $\delta = 1/2$. The adversary selects a random world $W$ at

the beginning. She selects world 0 with probability $1/2$ and world $w$ with probability $1/2T^{1-\beta}$ for all $w \in [T^{1-\beta}]$.

In world 0, we design the losses of experts as shown in Table 3. During the $w$-th interval with $w \in [T^{1-\beta}]$ being odd, we set $(\ell_{t,h_1}^{(1)}, \ell_{t,h_1}^{(2)}, \ell_{t,h_2}^{(1)}, \ell_{t,h_2}^{(2)}) = (0, c + \delta T^{\alpha-\beta}, 1, c - \delta T^{\alpha-\beta})$ for the first $T^\beta/2$ rounds and $(\ell_{t,h_1}^{(1)}, \ell_{t,h_1}^{(2)}, \ell_{t,h_2}^{(1)}, \ell_{t,h_2}^{(1)}) = (1, c, 0, c)$ for the second $T^\beta/2$ rounds. For $w$ being even, we swap the losses of the two experts, i.e., $(\ell_{t,h_1}^{(1)}, \ell_{t,h_2}^{(2)}, \ell_{t,h_2}^{(1)}, \ell_{t,h_2}^{(2)}) = (1, c - \delta T^{\alpha-\beta}, 0, c + \delta T^{\alpha-\beta})$ for the first $T^\beta/2$ rounds and $(\ell_{t,h_1}^{(2)}, \ell_{t,h_1}^{(2)}, \ell_{t,h_2}^{(1)}, \ell_{t,h_2}^{(2)}) = (0, c, 1, c)$ for the second $T^\beta/2$ rounds.

The intuition of constructing world $w \in [T^{1-\beta}]$ is described as below. In world $w$, the secondary loss is the same as that in world 0. The primary losses of each expert $h \in \mathcal{H}$ in the first $w - 1$ intervals are an approximately random permutation of that in world 0. Therefore, any algorithm will attain almost the same expected primary loss (around $(w-1)T^\beta/2$) in the first $w-1$ intervals of world $w$. The primary losses during the first $T^\beta/2$ rounds in the $w$-th interval are the same as those in world 0. Therefore, the cumulative losses from the beginning to any time $t$ in the first half of the $w$-th interval are almost the same in world 0 and world $w$, which makes the algorithm only dependent on the cumulative losses behave nearly the same during the first half of the $w$-th interval in two worlds. For $t = (w - 1/2)T^\beta + 1, \ldots, T$, we set $\ell_{t,h}^{(1)} = 1$ for all $h \in \mathcal{H}$, which indicates that any algorithms are unable to improve their primary loss after $t = (w - 1/2)T^\beta + 1$. To prove the theorem, we show that if the algorithm selects expert $h$ with loss $(1, c - \delta T^{\alpha-\beta})$ during the first half of the $w$-th interval with large fraction, then $\mathrm{Reg}^{(1)}$ will be large in world $w$; otherwise, $\mathrm{Reg}_c^{(2)}$ will be large in world 0.

More specifically, for the first $w - 1$ intervals in world $w$, we need to make the cumulative primary losses to be $(w-1)T^\beta/2$ with high probability. Let $t' = (w-1)T^\beta - 2\sqrt{(w-1)T^\beta \log(T)}$. For $t = 1, \ldots, t'$, $\ell_{t,h}^{(1)}$ are i.i.d. samples from $\mathrm{Ber}(1/2)$ for all $h \in \mathcal{H}$. We denote by $E_h^{(w)}$ the event of $\left| \sum_{t=1}^{t'} (\ell_{t,h}^{(1)} - 1/2) \right| \leq \sqrt{(w-1)T^\beta \log(T)}$ and denote by $E$ the event of $\cap_{h \in \mathcal{H}, w \in [T^{1-\beta}]} E_h^{(w)}$. If $E_{h_1}^{(w)} \cap E_{h_2}^{(w)}$ holds, we compensate the cumulative primary losses by assigning $\ell_{t,h}^{(1)} = 1$ for $(w-1)T^\beta/2 - \sum_{t=1}^{t'} \ell_{t,h}^{(1)}$ rounds and $\ell_{t,h}^{(1)} = 0$ for the remaining rounds during $t = t' + 1, \ldots, (w-1)T^\beta$ for all $h \in \mathcal{H}$ such that the cumulative primary losses in the first $w - 1$ intervals for both experts are $(w-1)T^\beta/2$ ; otherwise, we set $\ell_{t,h}^{(1)} = 1$ for all $h \in \mathcal{H}$ during $t = t' + 1, \ldots, (w-1)T^\beta$. Hence, if $E_{h_1}^{(w)} \cap E_{h_2}^{(w)}$, the cumulative losses $L_{(w-1)T^\beta,h}^{(1)} = (w-1)T^\beta/2$ for all $h \in \mathcal{H}$. To make it clearer, the values of the secondary losses in world $w$ for an even $w$ if $E_{h_1}^{(w)} \cap E_{h_2}^{(w)}$ holds are illustrated in Table 4.

Let $q_w = 2 \sum_{t=(w-1)T^\beta+1}^{(w-1/2)T^\beta} \mathbb{E}\left[ \mathbb{1}(\ell_{t,\mathcal{A}_t}^{(1)} = 0) \right] / T^\beta$ denote the expected fraction of selecting the expert with losses $(0, c + \delta T^{\alpha-\beta})$ in $w$-th interval in world 0 as well as that in world $w$ when $E$ holds. We denote by $\mathrm{Reg}^{(1,w)} = L_{T,\mathcal{A}}^{(1,w)} - L_{T,h_0}^{(1,w)}$ and $\mathrm{Reg}_{t'}^{(1,w)} = L_{t',\mathcal{A}}^{(1,w)} - L_{t',h_0}^{(1,w)}$ with $h_0 = \arg\min_{h \in \mathcal{H}} L_{T,h}^{(1,w)}$ being the best expert in hindsight the regret with respect to the primary loss for all times and the regret incurred during $t = 1, \ldots, t'$ in world $w$. We denote by $\mathrm{Reg}_c^{(2,w)}$ the regret to $cT$ with respect to the secondary loss in world $w$. Then we have

$$\mathbb{E}\left[ \mathrm{Reg}_c^{(2,W)} \mid W = 0 \right] = \sum_{w \in [T^{1-\beta}]} \frac{\delta(2q_w - 1)T^\alpha}{2} ,$$

and for all $w \in [T^{1-\beta}]$,

$$\mathbb{E}\left[ \mathrm{Reg}^{(1,W)} \mid W = w, E \right] \geq (1 - q_w)\frac{T^\beta}{2} + \mathbb{E}\left[ \mathrm{Reg}_{t'}^{(1,w)} \mid W = w, E \right] - \left( (w-1)T^\beta - t' \right)$$

$$\geq (1 - q_w)\frac{T^\beta}{2} - 2\sqrt{(w-1)T^\beta \log(T)} .$$

Due to Hoeffding's inequality and union bound, we have $\mathbb{P}\left[ \neg E_h^{(w)} \right] \leq \frac{2}{T^2}$ for all $h \in \mathcal{H}$ and $w \in [T^{1-\beta}]$ and $\mathbb{P}[\neg E] \leq \frac{4}{T^{1+\beta}}$. Let $Q = \frac{\sum_{w=1}^{T^{1-\beta}} q_w}{T^{1-\beta}}$ denote the average of $q_w$ over all $w \in [T^{1-\beta}]$. By taking expectation over the adversary, we have

$$\mathbb{E}\left[\max\left(\mathrm{Reg}^{(1)},\mathrm{Reg}_c^{(2)}\right)\right]$$

$$\geq \mathbb{P}\left[E\right]\cdot\mathbb{E}\left[\max\left(\mathrm{Reg}^{(1)},\mathrm{Reg}_c^{(2)}\right)|E\right]$$

$$\geq\left(1-\frac{4}{T^{1+\beta}}\right)\left(\frac{1}{2T^{1-\beta}}\sum_{w=1}^{T^{1-\beta}}\mathbb{E}\left[\mathrm{Reg}^{(1,W)}|W=w,E\right]+\frac{1}{2}\mathbb{E}\left[\mathrm{Reg}_c^{(2,W)}|W=0,E\right]\right)$$

$$\geq\frac{1}{2}\left(\frac{1}{2T^{1-\beta}}\left(\sum_w(1-q_w)\frac{T^\beta}{2}-2\sum_{w=1}^{T^{1-\beta}}\sqrt{(w-1)T^\beta\log(T)}\right)+\frac{\delta}{4}\sum_{w=1}^{T^{1-\beta}}(2q_w-1)T^\alpha\right)$$

$$\geq\frac{1}{8}(1-Q)T^\beta-\sqrt{T\log(T)}+\frac{\delta}{8}(2Q-1)T^{1-\beta+\alpha}$$

$$\geq\frac{1}{16}T^{\frac{1+\alpha}{2}}-\sqrt{T\log(T)}, \tag{3}$$

where Eq. (3) holds by setting $\beta=\frac{1+\alpha}{2}$ and $\delta=1/2$. $\qquad\square$

Table 3: The losses in world 0.

| experts\time | | $T^\beta/2$ | $T^\beta/2$ | $T^\beta/2$ | $T^\beta/2$ | $T^\beta/2$ | ... |
|---|---|---|---|---|---|---|---|
| $h_1$ | $\ell^{(1)}$ | 0 | 1 | 1 | 0 | 0 | ... |
| | $\ell^{(2)}$ | $c+\delta T^{\alpha-\beta}$ | $c$ | $c-\delta T^{\alpha-\beta}$ | $c$ | $c+\delta T^{\alpha-\beta}$ | ... |
| $h_2$ | $\ell^{(1)}$ | 1 | 0 | 0 | 1 | 1 | ... |
| | $\ell^{(2)}$ | $c-\delta T^{\alpha-\beta}$ | $c$ | $c+\delta T^{\alpha-\beta}$ | $c$ | $c-\delta T^{\alpha-\beta}$ | ... |

Table 4: The primary losses in world $w$ (which is even) if $E_{h_1}^{(w)}\cap E_{h_2}^{(w)}$ holds.

| experts\time | | $t'$ | $(w-1)T^\beta-t'$ | $T^\beta/2$ | $T^\beta/2$ | $T-T^\beta$ |
|---|---|---|---|---|---|---|
| $h_1$ | $\ell^{(1)}$ | i.i.d. from $\mathrm{Ber}(1/2)$ | compensate | 1 | 1 | 1 |
| $h_2$ | $\ell^{(1)}$ | i.i.d. from $\mathrm{Ber}(1/2)$ | compensate | 0 | 1 | 1 |

## C  Proof of Theorem 7

**Theorem 7.** *Let $T_{h^*}=\omega(T^\alpha)$ for all $h^*\in\mathcal{H}$. There exists an adversary such that any algorithm achieving $\mathrm{SleepReg}_{h^*}^{(1)}=o(T_{h^*})$ for all $h^*\in\mathcal{H}$ will incur $\mathrm{Reg}_c^{(2)}=\Omega(KT^\alpha)$ for $K=O(\log(T))$.*

*Proof.* The idea is to construct an example in which the best expert with respect to the primary loss is deactivated sequentially while incurring an extra $\Theta(T^\alpha)$ secondary loss. In the example, we set $\mathcal{H}=[K]$. Let $T_k=T^{\alpha+\frac{(k-1)(1-\alpha)}{K-1}}$ for $k\in[K]$ and $T_0=0$. For each expert $k\in\mathcal{H}$, we set $(\ell_{t,k}^{(1)},\ell_{t,k}^{(2)})=(1,c)$ for $t\leq T_{k-1}$ and $(\ell_{t,k}^{(1)},\ell_{t,k}^{(2)})=(0,c+\frac{\delta T^\alpha}{T_k-T_{k-1}})$ for $t\geq T_{k-1}+1$. Then expert $k$ will be deactivate at time $t=T_k$. For any algorithm with $\mathrm{SleepReg}_k^{(1)}=o(T_k)$ for all $k\in\mathcal{H}$, expert $k$ should be selected for $T_k-2T_{k-1}-o(T_k)$ rounds during $t=T_{k-1}+1,\ldots,T_k$. Therefore, we have $\mathrm{Reg}_c^{(2)}\geq\sum_{k\in[K]}\frac{\delta T^\alpha}{T_k-T_{k-1}}(T_k-2T_{k-1}-o(T_k))=\Omega(KT^\alpha)$. $\qquad\square$

## D  Proof of Theorem 8

**Theorem 8.** *Running Algorithm 2 can achieve*

$$\mathrm{SleepReg}^{(1)}(h^*)=O(\sqrt{\log(K)T^{1+\alpha}}+T_{h^*}\sqrt{\log(K)T^{\alpha-1}}),$$

*for all $h^* \in \mathcal{H}$ and*

$$\text{Reg}_c^{(2)} = O(\sqrt{\log(K)T^{1+\alpha}} + \log(K)T^\alpha N + NKT^\alpha) \,.$$

*Proof.* Let $\widetilde{L}_h^{(1,h^*)} = \sum_{m=1}^{T^{1-\alpha}} I_{h^*}(e_m)\widetilde{\ell}_{e_m,h}^{(1)}$ and $\widetilde{L}_{\mathcal{A}}^{(1,h^*)} = \sum_{m=1}^{T^{1-\alpha}} I_{h^*}(e_m)\widetilde{\ell}_{e_m,\mathcal{A}}^{(1)}$ denote the cumulative pseudo primary losses of expert $h$ and algorithm $\mathcal{A}$ during the time when $h^*$ is active. First, since we update $w_{m+1,h}^{h^*} = w_{m,h}^{h^*}\eta^{I_{h^*}(e_m)(\widetilde{\ell}_{e_m,h}^{(1)}-\eta\widetilde{\ell}_{e_m,\mathcal{A}}^{(1)})+1} \le w_{m,h}^{h^*}$ with $\eta \in [1/\sqrt{2},1]$ and experts will not be reactivated between (not including) $t_n$ and $t_{n+1}$, the probability of following the first rule on Line 7 in Algorithm 2, which is $\frac{w_{m+1,h_m}}{w_{m,h_m}}$, is legal. Then we show that at each epoch $m$, the probability of getting $h_m = h$ is $\mathbb{P}[h_m = h] = p_{m,h}$. The proof follows Lemma 1 by [Geulen et al., 2010]. For an reactivating epoch $m \in \{(t_n-1)/T^\alpha + 1\}_{n=0}^N$, $h_m$ is drawn from $p_m$ and thus, $\mathbb{P}[h_m = h] = p_{m,h}$ holds. For other epochs $m \notin \{(t_n-1)/T^\alpha + 1\}_{n=0}^N$, we prove it by induction. Assume that $\mathbb{P}[h_{m-1} = h] = p_{m-1,h}$, then

$$\mathbb{P}[h_m = h] = \mathbb{P}[h_{m-1} = h]\frac{w_{m,h}}{w_{m-1,h}} + p_{m,h}\sum_{h' \in \mathcal{H}}\mathbb{P}[h_{m-1} = h']\left(1 - \frac{w_{m,h'}}{w_{m-1,h'}}\right)$$

$$= \frac{w_{m-1,h}}{W_{m-1}} \cdot \frac{w_{m,h}}{w_{m-1,h}} + \frac{w_{m,h}}{W_m}\left(1 - \sum_{h' \in \mathcal{H}}\frac{w_{m-1,h'}}{W_{m-1}} \cdot \frac{w_{m,h'}}{w_{m-1,h'}}\right)$$

$$= p_{m,h} \,.$$

To prove the upper bound on sleeping regrets, we follow Claim 12 by Blum and Mansour [2007] to show that $\sum_{h,h^*} w_{m,h}^{h^*} \le K\eta^{m-1}$ for all $m \in [T^{1-\alpha}]$.

First, we have

$$W_m\widetilde{\ell}_{e_m,\mathcal{A}}^{(1)} = W_m\sum_{h \in \mathcal{H}}p_{m,h}\widetilde{\ell}_{e_m,h}^{(1)} = \sum_{h \in \mathcal{H}}w_{m,h}\widetilde{\ell}_{e_m,h}^{(1)} = \sum_{h \in \mathcal{H}}\sum_{h^* \in \mathcal{H}}I_{h^*}(e_m)w_{m,h}^{h^*}\widetilde{\ell}_{e_m,h}^{(1)} \,. \quad (4)$$

Then according to the definition of $w_{m,h}^{h^*}$, we have

$$\sum_{h \in \mathcal{H}, h^* \in \mathcal{H}}w_{m+1,h}^{h^*}$$

$$= \sum_{h \in \mathcal{H}, h^* \in \mathcal{H}}w_{m,h}^{h^*}\eta^{I_{h^*}(e_m)(\widetilde{\ell}_{e_m,h}^{(1)}-\eta\widetilde{\ell}_{e_m,\mathcal{A}}^{(1)})+1}$$

$$\le \eta\left(\sum_{h \in \mathcal{H}, h^* \in \mathcal{H}}w_{m,h}^{h^*}\left(1 - (1-\eta)I_{h^*}(e_m)\widetilde{\ell}_{e_m,h}^{(1)}\right)\left(1 + (1-\eta)I_{h^*}(e_m)\widetilde{\ell}_{e_m,\mathcal{A}}^{(1)}\right)\right)$$

$$\le \eta\left(\sum_{h \in \mathcal{H}, h^* \in \mathcal{H}}w_{m,h}^{h^*} - (1-\eta)\left(\sum_{h \in \mathcal{H}, h^* \in \mathcal{H}}w_{m,h}^{h^*}I_{h^*}(e_m)\widetilde{\ell}_{e_m,h}^{(1)} - W_m\widetilde{\ell}_{e_m,\mathcal{A}}^{(1)}\right)\right)$$

$$= \eta\sum_{h \in \mathcal{H}, h^* \in \mathcal{H}}w_{m,h}^{h^*} \,,$$

where the last inequality adopts Eq. (4). Combined with $w_{1,h}^{h^*} = \frac{1}{K}$ for all $h \in \mathcal{H}, h^* \in \mathcal{H}$, we have $\sum_{h,h^*} w_{m+1,h}^{h^*} \le K\eta^m$. Since $w_{m+1,h}^{h^*} = w_{1,h}^{h^*}\eta^{\sum_{i=1}^m I_{h^*}(e_i)\widetilde{\ell}_{e_i,h}^{(1)}-\eta\sum_{i=1}^m I_{h^*}(e_i)\widetilde{\ell}_{e_i,\mathcal{A}}^{(1)}+m} \le K\eta^m$, we have

$$\widetilde{L}_{\mathcal{A}}^{(1,h^*)} - \widetilde{L}_h^{(1,h^*)} \le \frac{(1-\eta)\widetilde{L}_h^{(1,h^*)} + \frac{2\log(K)}{\log(1/\eta)}}{\eta} \,.$$

By setting $\eta = 1 - \sqrt{2\log(K)/T^{1-\alpha}}$, we have $\text{SleepReg}^{(1)}(h^*) \le 2\sqrt{\log(K)T^{1+\alpha}} + 2T_{h^*}\sqrt{\log(K)T^{\alpha-1}}$.

To derive $\text{Reg}_c^{(2)}$, we bound the number of switching times. We denote by $S_n$ the number of epochs in which some experts are deactivated during $(t_n-1)/T^\alpha + 1 < m < (t_{n+1}-1)/T^\alpha + 1$ and by

$\tau_1, \ldots, \tau_{S_n}$ the deactivating epochs, i.e., $\Delta\mathcal{H}_{\tau_i} \neq \emptyset$ for $i \in [S_n]$. We denote by $\alpha_m$ the probability of following the second rule at line 7 in Algorithm 2, which is getting $h_m$ from $p_m$. Then we have

$$\alpha_m = \sum_{h \in \mathcal{H}} \mathbb{P}\left[h_{m-1} = h\right]\left(1 - \frac{w_{m,h}}{w_{m-1,h}}\right) = \sum_{h \in \mathcal{H}} \frac{w_{m-1,h}}{W_{m-1}}\left(1 - \frac{w_{m,h}}{w_{m-1,h}}\right) = \frac{W_{m-1} - W_m}{W_{m-1}}\ .$$

Since $W_{\tau_{i+1}}/W_{\tau_i+1} \geq \eta^{2(\tau_{i+1} - \tau_i - 1)}$, we have

$$\sum_{m=\tau_i+1}^{\tau_{i+1}} \alpha_m \leq 1 - \sum_{m=\tau_i+2}^{\tau_{i+1}} \log(1 - \alpha_m) = 1 - \sum_{m=\tau_i+2}^{\tau_{i+1}} \log\left(\frac{W_m}{W_{m-1}}\right) = 1 + \log\left(\frac{W_{\tau_{i+1}}}{W_{\tau_i+1}}\right)$$

$$\leq 1 + 2\sqrt{2}(\tau_{i+1} - \tau_i - 1)(1 - \eta) = 1 + 4(\tau_{i+1} - \tau_i - 1)\sqrt{\log(K)/T^{1-\alpha}}\ .$$

Therefore, during time $(t_n - 1)/T^\alpha \leq m < (t_{n+1} - 1)/T^\alpha$, the algorithm will switch at most $K + 4(t_{n+1} - t_n)\sqrt{\log(K)/T^{1-\alpha}} + 1$ times in expectation, which results in $\mathrm{Reg}_c^{(2)} \leq 4\delta\sqrt{\log(K)T^{1+\alpha}} + \delta N(K+1)T^\alpha = O(\sqrt{\log(K)T^{1+\alpha}} + NKT^\alpha)$  □