[Reviews · NeurIPS 2020]

Review 1

Summary and Contributions: The authors analyze the online learning setting with a bicriteria of primary loss and secondary losses. They give a series of results:  1)  No-regret guarantees for primary loss while performing no worse than the worst expert for secondary loss is impossible in general. 2) present algorithms and (upper and lower) regret guarantees for  “good” scenarios in which all experts satisfy bounded variance assumption 3) present algorithms and (upper and lower) regret guarantees for “bad” scenario in which experts partially satisfy bounded variance assumption and access to an external oracle to deactivate and reactivate experts. 

Strengths: The authors present a very thorough analysis of the setting of learning with primary and secondary losses. They delve into different aspects of the problem and find solutions under the hardness constraints of the setting. 

Weaknesses: I found it strange to analyze a type of sleeping expert setting with an oracle that deactivates experts which basically make the problem hard. I am not sure how realistic this oracle is and how interesting this part of the paper is to the community.  There is a missing reference to sleeping experts literature -- Kleinberg, Robert, Niculescu-Mizil, Alexandru, andSharma, Yogeshwer. Regret bounds for sleeping expertsand bandits. Machine Learning, 2008.  Is there a reason the authors used the sleeping expert definition in the paper instead the one in Kleinberg? I did not understand their algorithm in section 4.2 "at the beginning of each epoch, we apply L over the average primary loss of each epoch." The paper is very dense and at times hard to follow. 

Correctness: The claims appear correct, but I did not check proofs.

Clarity: The paper is well written, but too dense at times.

Relation to Prior Work: Mostly except some references to sleeping experts literature.

Reproducibility: Yes

Additional Feedback:


Review 2

Summary and Contributions: This paper introduces a new learning-with-expert-advice setting where in each round the learner receives a primary loss vector and a secondary loss vector simultaneously. The goal is to achieve low regret wrt the primary loss, while minimizing the regret to a linear threshold with respect to the secondary loss. The authors firstly show that, without further assumptions, it is impossible to achieve sublinear regret with respect to the two losses simultaneously. They then consider two scenarios where meaningful results can be achieved. The first scenario is when the variance in any time interval of each expert is bounded, and the second scenario is that some variance-related oracles are accessible. For the two settings, the authors proposed two algorithms respectively which can achieve sublinear regret bounds.

Strengths: 1. The problem is interesting and well-motivated. Learning with primary and secondary losses widely exists in real-world applications such as hiring, advertising, but it is rarely considered in online learning. Although there are some previous work consider muliti-objective optimization in online learning, e.g. [Sani et al. 2014;Auer et al., 2016], they can not cover the specific bicriteria setting considered in this paper. 2. This paper conducts a thorough theoretical analysis for the proposed problem. They first prove an impossibility result for the general setting by an counter example, and then provide several meaningful results (algorithm with sublinear regret) under assumptions that the secondary losses of the experts are bounded.

Weaknesses: Algorithms 1 and 2 require the parameter alpha as input, which is related to the upper bound of variance of the experts’ secondary losses. However, in general, this value can only be obtained after the whole learning process. --------------------------------------Post Rebuttal--------------------------- The authors have cleared my concerns in the rebuttal. I am happy to raise my score.

Correctness: I have read the main paper and made high level checks of the proofs, and I didn’t find any significant errors.

Clarity: The paper is generally well-written and structured clearly.

Relation to Prior Work: The relation to prior work is clearly discussed in general. I noticed that there are some other work in (bandit) online learning such as [Turgay et al., 2018] also considered the multi-objective setting. It would be better if the authors could include these work in Section 1.2. Turgay, E., Oner, D., & Tekin, C. Multi-objective Contextual Bandit Problem with Similarity Information. In AISTATS2018.

Reproducibility: Yes

Additional Feedback:


Review 3

Summary and Contributions: The paper addresses an old problem of predicting w.r.t. two loss functions. It is an interesting and old prediction with expert advice problem, which has been intermittently addressed in different contexts without much success. The setup of the paper is as follows. There are K experts and they get (bounded) losses \ell_{t,1}^1,...,\ell_{t,K}^1 and \ell_{t,1}^2,...,\ell_{t,K}^2. The learner must produce a vector of weights p_{t,1},...,p_{t,K} and, when the experts' losses are revealed, it suffers losses \sum_k p_{t,k}\ell_{t,k}^1 and \sum_k p_{t,k}\ell_{t,k}^2. It is important that there is only one vector of weights. The losses sum to L_T^1 and L_T^2 and the corresponding regrets can be evaluated. It is easy to show that under the general assumptions \max(R_t^1,R_t^2) = \Omega(t). The contribution of the paper is as follows. If one of the losses grows sublinearly on subintervals, then sublinear regret can be obtained (growth T^\alpha translates to regret \sqrt{T^{1+\alpha}}). The paper also covers the specialist experts scenario of Freund et al: experts may sleep, i.e., refrain from prediction, i.e., some losses may be missing. Then the weights are renormalised over the awake experts. This is a nice addition and it is good to know that this extension is supported by the algorithm of the paper, but this is not a core result. The paper then proceeds to consider the scenario when we get oracle information of a deterioration of an expert's performance w.r.t. a loss function so that that expert could be disable. This seems to be an interesting and practical scenario.

Strengths: The results of the paper appear very interesting and important for the prediction with expert advice community. The scenario and the algorithm are obviously practical so interesting applications may follow.

Weaknesses: One cannot help noticing that the regret term in the results is quite large. An o(T) bound does not look impressive. I think this is inevitable under the circumstances. There might be different conditions leading to better regret, but they are yet to be discovered.

Correctness: Yes

Clarity: Yes

Relation to Prior Work: Yes. The paper arXiv:0902.4127 actually considers the scenario of prediction w.r.t. multiple loss functions as in this paper. Citation (Chernov and Vovk, 2009) covers a different scenario as correctly explained in the paper. However, arXiv:0902.4127 is a longer version and it considers exactly the scenario of multiple losses in the section 'Multiobjective prediction with expert advice'. The result is not very general though because it imposes heavy geometric restrictions on loss functions. No, I am not one of the authors, just happen to know this.

Reproducibility: Yes

Additional Feedback: Thank you for the answer to my comments. I am keeping my score.

[Author Response · NeurIPS 2020]

We would like to thank the reviewers for their thoughtful comments. We address below the main questions.

Reviewer #1

1. How realistic is the oracle of deactivating experts and how interesting is this part to the community?

Our intent in introducing the oracle is just to provide a formal way to generalize from "assuming that all
experts have low variance" to "competing with the best low-variance expert" or "competing with the best
expert in the period in which it has low variance". We will try to be more clear about that.

2. Why using the sleeping expert definition in the paper instead of the one in Kleinberg et al.?

Kleinberg et al. study the regret to the best ordering of experts, which is indeed different from our regret
definition. Kanade and Steinke [2014] show that achieving optimal regret bound in this setting is computa-
tionally hard. Computationally efficient no-regret algorithms (e.g., Blum and Mansour [2007]) are known in
our setting. Tackling the computation hardness of the sleeping setting is not the focus of this paper. We will
add the reference and more discussion on the sleeping settings.

3. The meaning of "at the beginning of each epoch, we apply L over the average primary loss of each epoch"?

The algorithm in Section 4.2 divides $T$ into $T^{1-\alpha}$ epochs evenly. Let $e_i = \{(i-1)T^\alpha + 1, \ldots, iT^\alpha\}$ denote
the $i$-th epoch and $\ell^{(1)}_{e_i,h} = \sum_{t \in e_i} \ell^{(1)}_{t,h}/T^\alpha$ denote the average primary loss of the $i$-th epoch. We run $\mathcal{L}$ over
$\{\ell^{(1)}_{e_i,h}\}_{h \in \mathcal{H}}$ for $i = 1, \ldots, T^{1-\alpha}$. We will make sure to specify the algorithm.

Reviewer #2

1. The parameter $\alpha$ is required as input, whose value can only be obtained after the learning process?

It is true that our algorithms need some information of $\alpha$ to obtain meaningful regret bounds. However, our
algorithms only uses $\alpha$ to determine the length of epochs. Our algorithms can run without $\alpha$ with a more
complicated theoretical guarantee. For example, we run Algorithm 1 with $\mathcal{A}_{\mathrm{SL}}(\mathcal{L})$ running $\mathcal{L}$ over $T^{1-\beta}$
epochs instead of $T^{1-\alpha}$ epochs, where $\beta$ is given as input. Then Algorithm 1 can achieve $\mathrm{SleepReg}^{(1)}(h^*) =$
$O(\sqrt{T_{h^*}T^\beta})$ and $\mathrm{Reg}_c^{(2)} \leq \delta T^\alpha(\sqrt{\log(K)T^{1-\beta}} + K) = O(\sqrt{T^{1+2\alpha-\beta}})$. To make the bounds meaningful,
we need to set $2\alpha - 1 < \beta \leq \alpha$ (assuming $T_{h^*} = \omega(T^\alpha)$ as mentioned in Section 5.1). If $\alpha \leq 1/2$, we can
set $\beta$ to any value in $[0, \alpha]$. Therefore, we do not need the exact value of $\alpha$ as input.

2. Missing reference?

We will add the reference.

Reviewer #3

1. The regret $o(T)$ is large?

We agree that $o(T)$ is inevitable when $\alpha$ is close to 1 as shown in the lower bounds (Theorem 2 & 3) in the
"good" scenario. In the "bad" scenario, when $T_{h^*}$ is close to $T^\alpha$, the sleeping regret $o(T_{h^*})$ is inevitable as
we mention in the paper.

## References

Avrim Blum and Yishay Mansour. From external to internal regret. *Journal of Machine Learning Research*, 8(Jun):
1307–1324, 2007.

Varun Kanade and Thomas Steinke. Learning hurdles for sleeping experts. *ACM Transactions on Computation Theory
(TOCT)*, 6(3):1–16, 2014.


[Meta-Review · NeurIPS 2020]

There were three reviews recommending accepting, but minor issues were raised regarding relation to previous work and the details of the algorithm. The reply by the authors satisfied the reviewers, and after discussion, all the reviewers are clearly in support of accepting.